# Facilitating Adaptive Forest Management under Climate Change: A Spatially Specific Synthesis of 125 Species for Habitat Changes and Assisted Migration over the Eastern United States

**Louis R. Iverson [1,*], Anantha M. Prasad [1], Matthew P. Peters [1] and Stephen N. Matthews [1,2]**

[1]  USDA Forest Service, Northern Research Station, Northern Institute of Applied Climate Science, 359 Main Road, Delaware, OH 43015, USA; anantha.prasad@usda.gov (A.M.P.); matthew.p.peters@usda.gov (M.P.P.); matthews.204@osu.edu (S.N.M.)

[2]  School of Environment and Natural Resources, Kottman Hall, 2021 Coffey Road, The Ohio State University, Columbus, OH 43210, USA

*  Correspondence: louis.iverson@usda.gov; Tel.: +1-740-368-0097

**Abstract:** We modeled and combined outputs for 125 tree species for the eastern United States, using habitat suitability and colonization potential models along with an evaluation of adaptation traits. These outputs allowed, for the first time, the compilation of tree species' current and future potential for each unit of 55 national forests and grasslands and 469 1 × 1 degree grids across the eastern United States. A habitat suitability model, a migration simulation model, and an assessment based on biological and disturbance factors were used with United States Forest Service Forest Inventory and Analysis data to evaluate species potential to migrate or infill naturally into suitable habitats over the next 100 years. We describe a suite of variables, by species, for each unique geographic unit, packaged as summary tables describing current abundance, potential future change in suitable habitat, adaptability, and capability to cope with the changing climate, and colonization likelihood over 100 years. This resulting synthesis and summation effort, culminating over two decades of work, provides a detailed data set that incorporates habitat quality, land cover, and dispersal potential, spatially constrained, for nearly all the tree species of the eastern United States. These tables and maps provide an estimate of potential species trends out 100 years, intended to deliver managers and publics with practical tools to reduce the vast set of decisions before them as they proactively manage tree species in the face of climate change.

**Keywords:** suitable habitat; migration; dispersal model; range shifts; decision-support tools; adaptive management; DISTRIB-II; SHIFT

## 1. Introduction

Human-induced rapid climate change will have profound impacts [1,2], including large impacts on the earth's biodiversity [3]. A critical need towards better understanding potential impacts and adaptation strategies for biodiversity conservation hinges on the development of good predictive models, yet basic biological information, especially related to species interactions, dispersal, demography, physiology, and evolution, is sorely lacking for most of the earth's biodiversity [4]. Species distribution models, by themselves, often do not adhere to consistent standards for model building, biological data incorporation, and model evaluation [5], and do not adequately account for these biological attributes [6]. Even for well-studied species, critical information is often lacking for the development of more realistic models, though such models are beginning to emerge for selected

species that provide more realism via hybrid models that combine aspects of species distribution models and process models, and that can assist in forest management [7,8]. However, tools that help manage the entire suite of species in the face of a changing climate are vital [9].

Our objective was to move beyond the individual evaluation of tree species or forest types by synthesizing the disparate species-based results and tying them to unique geographic units, then providing a comprehensive report of current and potential future configurations of forest communities impacted by climate change. In doing so, we are likely to gain new understanding of both the current configuration of forest communities but also be in a more robust position to evaluate how climate change will continue to exert macroscale pressures across the continent. Our approach has been to combine the results of a habitat suitability model (DISTRIB-II, for projecting potential future suitable habitats), a probabilistic model (SHIFT, for estimating natural migration into projected suitable habitat within 100 years), a literature-based set of modification factors (ModFacs, for evaluating traits associated with adaptation), and a recent assessment of Forest Inventory and Analysis data (FIA, www.fs.fed.us/fia) [10,11].

We use two spatial subdivisions of the eastern US in this approach to assess the potentials of the tree species to resist, adapt, or migrate throughout this century—469 units comprising a grid of $1 \times 1$ degrees of latitude and longitude, and 55 units comprising the eastern US' national forests and grasslands. Within each unit, we evaluate many aspects of tree species' potential behavior in light of the changing climate and tabulate them in an information-packed table. Within each table, a user will be able to assess the dynamics of the forest species in their unit, both currently as well as potentially out to century's end. We focus on evaluating species for their capacities to persist or migrate in the changing climate. By persist, we mean a species' capability to resist or adapt to the added stresses associated with the changing climate, based on a species abundance and innate attributes (i.e., traits). Because our models cannot detect which mechanism (resist or adapt) is in play to allow the species to persist, we generalize to infer where conditions may allow the species to persist into the future. With migration, we mean a species' tendency to migrate (most often northward) to follow suitable climate conditions into the future under the changed climate, and we do estimate migration with our models. Though migration can also mean movement of species upslope, we cannot address that in this study due to the coarse level of analyses.

With this combined species approach, our objective is to focus on providing tables of information for any location in the eastern US, along with any eastern national forest or grassland. This combination of the species distribution modeling of DISTRIB-II and the migration simulations of SHIFT allow, for the first time, a credible picture of what tree species conditions are like now as well as potential species changes into the future. The intention is to provide guidelines and tools enabling managers and publics alike to engage proactively in adaptation efforts in the face of the changing climate. With this information aggregated spatially, we are also able to map spatial trends of any of the tabulated data from the $1 \times 1$ degree units or national forests and grasslands within the eastern US. This mapping approach allows a comprehensive evaluation of a full suite of tree species and their spatial trends, both currently and potentially into the future.

The intention is to create a set of outputs geared to assist in decision support for local and regional forest management in the face of a changing climate. In keeping with our earlier products associated with the Northern Institute of Applied Climate Science [12–18], this undertaking is an ambitious step forward: to provide decision tools to managers by comprehensively synthesizing the habitat and colonization potential of 125 tree species in the eastern United States.

## 2. Material and Methods

### 2.1. Study Area

This study encompasses the United States east of the 100th meridian. To summarize model outputs, the study region was sliced into 469 $1 \times 1$ degree grids (hereafter $1 \times 1$), which provides a

wall-to-wall coverage for any location in the region to assess general trends at a coarsely summarized scale (Figure 1). In addition, we subset only the 55 national forests and grasslands (hereafter NF) to exemplify the capability to assess any set of geographic entities within the region (Figure 1). These two spatial units provide unique geographic footprints to (1) provide a continuous grid where transitions and patterns in species can be quantified; and (2) the federal lands provide a spatially distributed set of focal sites where differences and similarities can be evaluated from a management perspective.

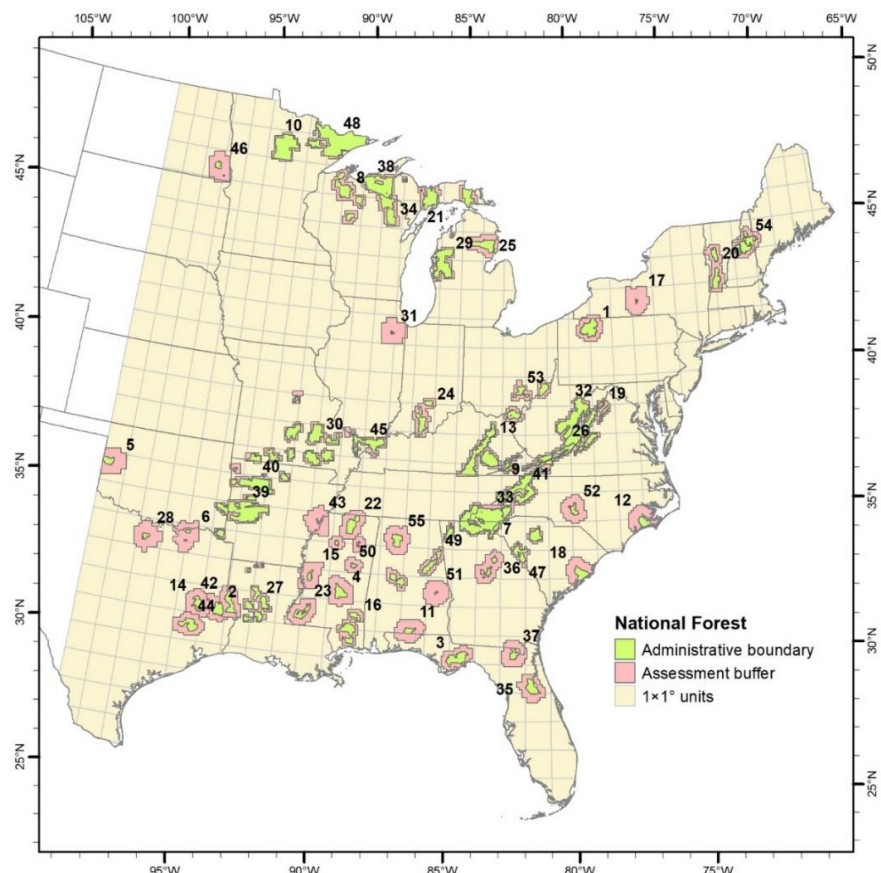

**Figure 1.** The National Forests and Grasslands (NF), superimposed on the 1 × 1 degree grid of the eastern US. Those NF locations occupying <8000 km$^2$ were buffered to attain at least that size, in order to access sufficient FIA plots for reasonable model outputs. The names of the forests or grasslands to which the numbers correspond are presented in Table 3. The 469 1 × 1 degree units are also depicted and provide a wall-to-wall coverage. The individual files are named by their southeast boundary, such that, for example, a GPS coordinate of 42.44° N latitude, −82.55° W longitude will be named S42_E82.

## 2.2. The Synthetic Approach

The overall synthesizing approach uses the results of the previously mentioned DISTRIB-II, SHIFT, and ModFacs model components; details on these components are provided in an online supplement as well as previous papers (Figure 2, Supplementary File 1—Climate Change Atlas description) [11,19–21]. DISTRIB-II provides estimates of habitat suitability (or habitat quality, HQ), by species, to century's end according to various scenarios of climate change. A hybrid lattice of 20 × 20 or 10 × 10 km cells was used for DISTRIB-II modeling with the cell size determined according to the number of forested FIA plots in each cell. SHIFT estimates colonization likelihood, CL into new habitats, for each 1 × 1 km cell over approximately 100 years, by running the algorithm 100 times, with each time a cell gets colonized counting as a 1% probability of colonization. ModFacs uses a literature approach to assess 12 disturbance and 9 biological attributes of species towards their adaptability (Adap) to deal with additional stresses likely under climate change.

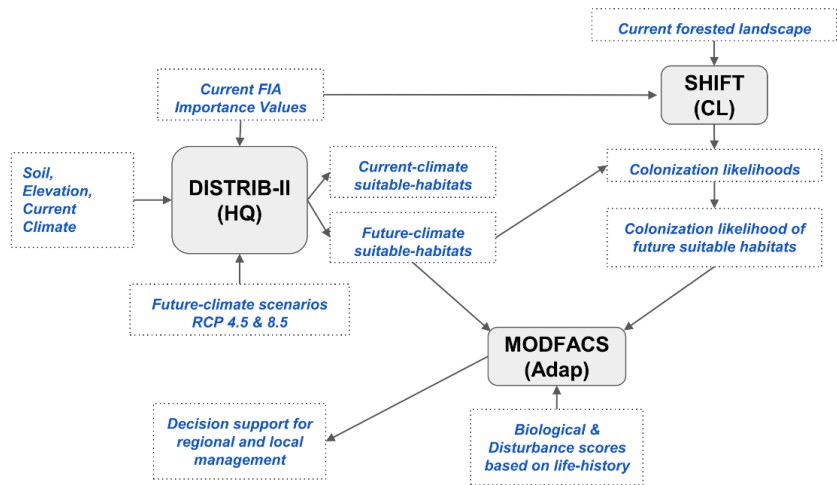

**Figure 2.** Flowchart depicting workflow to produce the DISTRIB-II and SHIFT outputs. DISTRIB-II predicts current and future habitat suitability/quality (HQ) and SHIFT calculates colonization likelihoods (CL) and ModFacs estimates adaptability (Adap).

A focus of this paper is a tabular output for a specific 1 × 1 or NF unit that provides species-level details on current status, potential future status, and features related to possible management such as assisted migration; the outputs are derived from combinations of the three model components (Figure 3). Tables, such as that produced in Microsoft Excel for the Allegheny NF (Figure 4, Supplementary File 2—Allegheny table) and identical to those produced for the 1 × 1 degree units, use the headers described below (in bold print) but also include, in seven accompanying excel sheets, a much more inclusive set of data along with suggestions on how to interpret the tables, variable descriptions, and a set of questions that can be asked of the data. A full explanation of the table components is also provided in Supplementary File 3—Explanation of species tables. These tables were generated for each of the 55 NF units and the 469 1 × 1 units. These tables are available for download at https://doi.org/10.2737/Climate-Change-Atlas-Combined-v4.

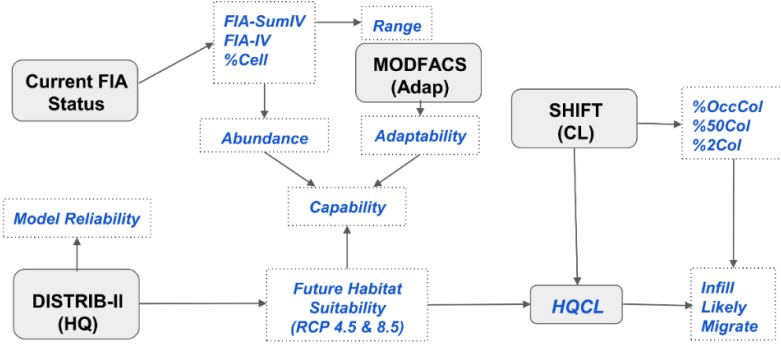

**Figure 3.** Flowchart depicting workflow for outputs of DISTRIB-II (habitat quality, HQ) and SHIFT (colonization likelihood, CL) to produce capability and candidate species for Infill (the number of species that can be considered likely to expand under a RCP, and thus higher candidates for infill planting), Likely (the species was not found in FIA plots, but there is a relatively high probability that the species exists in the unit), or Migrate (the number of species with at least some chance of migrating naturally into each unit, and thus higher candidates for artificial migration). HQCL is the combination of habitat quality (HQ) and colonization likelihood (CL). %OccCol pertains to the percentage of the unit that is either already occupied or with at least a 50% probability of becoming potentially occupied (according to SHIFT) within 100 years. %50Col (or %2Col) pertains to the percentage of the unit not already occupied that has at least a 50% (or 2%) probability of getting colonized within 100 years.

Allegheny

**National Forests and Grasslands**
**Climate Change Atlas Tree Species**
**Current and Potential Future Habitat, Capability, and Migration**

| Common Name | Scientific Name | Range | MR | %Cell | FIAsum | FIAiv | ChngCl45 | ChngCl85 | Adap | Abund | Capabil45 | Capabil85 | SHIFT45 | SHIFT85 | SSO | N |
|---|---|---|---|---|---|---|---|---|---|---|---|---|---|---|---|---|
| red maple | Acer rubrum | WDH | High | 98.8 | 2598.07 | 26.3 | Sm. dec. | Lg. dec. | High | Abundant | Good | Good | | | 1 | 1 |
| black cherry | Prunus serotina | WDL | Medium | 92.7 | 1609.09 | 17.36 | Sm. dec. | Sm. dec. | Low | Abundant | Good | Good | | | 1 | 2 |
| sugar maple | Acer saccharum | WDH | High | 84.1 | 933.7 | 11.1 | Sm. dec. | Sm. dec. | High | Abundant | Good | Good | | | 1 | 3 |
| American beech | Fagus grandifolia | WDH | High | 86.6 | 792.88 | 9.16 | Lg. dec. | Lg. dec. | Medium | Abundant | Good | Good | | | 1 | 4 |
| eastern hemlock | Tsuga canadensis | NSH | High | 68.3 | 721.04 | 10.56 | Lg. dec. | Lg. dec. | Low | Abundant | Good | Good | | | 1 | 5 |
| sweet birch | Betula lenta | NDH | High | 87.8 | 569.73 | 6.49 | Sm. dec. | Lg. dec. | Low | Abundant | Good | Good | | | 1 | 6 |
| northern red oak | Quercus rubra | WDH | Medium | 58.5 | 490.73 | 8.38 | Sm. inc. | Sm. inc. | High | Common | Very Good | Very Good | | | 1 | 7 |
| white oak | Quercus alba | WDH | Medium | 39 | 311.79 | 7.99 | Lg. inc. | Lg. inc. | High | Common | Very Good | Very Good | | | 1 | 8 |
| white ash | Fraxinus americana | WDL | Medium | 57.3 | 239.06 | 4.17 | Sm. inc. | Sm. inc. | Low | Common | Very Good | Very Good | | | 1 | 9 |
| yellow birch | Betula alleghaniensis | NDL | High | 63.4 | 217.91 | 3.44 | Lg. dec. | Lg. dec. | Medium | Common | Fair | Fair | | | 1 | 10 |
| pin cherry | Prunus pensylvanica | NSL | Low | 28 | 132.29 | 4.72 | Lg. dec. | Very Lg. dec. | Medium | Common | Fair | Lost | | | 1 | 11 |
| serviceberry | Amelanchier spp. | NSL | Low | 50 | 112.26 | 2.25 | Lg. dec. | Lg. dec. | Medium | Common | Fair | Fair | | | 1 | 12 |
| quaking aspen | Populus tremuloides | WDH | High | 20.7 | 85.9 | 4.14 | Lg. dec. | Very Lg. dec. | Medium | Common | Fair | Lost | | | 1 | 13 |
| black oak | Quercus velutina | WDH | High | 23.2 | 84.56 | 3.65 | Lg. inc. | Lg. inc. | Medium | Common | Very Good | Very Good | | | 1 | 14 |
| eastern white pine | Pinus strobus | WDH | High | 23.2 | 83 | 3.58 | Lg. inc. | Lg. inc. | Low | Common | Very Good | Very Good | | | 1 | 15 |
| American hornbeam; muscle | Carpinus caroliniana | WSL | Low | 39 | 74.21 | 1.9 | Lg. dec. | No change | Medium | Common | Fair | Good | | | 1 | 16 |
| chestnut oak | Quercus prinus | NDH | High | 22 | 73.7 | 3.36 | Lg. inc. | Lg. inc. | High | Common | Very Good | Very Good | | | 1 | 17 |
| bigtooth aspen | Populus grandidentata | NSL | Medium | 20.7 | 66.98 | 3.23 | Sm. inc. | Sm. dec. | Medium | Common | Very Good | Fair | | | 1 | 18 |
| cucumbertree | Magnolia acuminata | NSL | Low | 36.6 | 63.74 | 1.74 | No change | Sm. dec. | Medium | Common | Good | Fair | | | 1 | 19 |
| shagbark hickory | Carya ovata | WSL | Medium | 7.3 | 61.68 | 8.43 | No change | No change | Medium | Common | Good | Good | Infill ++ | Infill ++ | 1 | 20 |
| American basswood | Tilia americana | WSL | Medium | 23.2 | 52.45 | 2.26 | Lg. inc. | Lg. inc. | Medium | Common | Very Good | Very Good | | | 1 | 21 |
| yellow-poplar | Liriodendron tulipifera | WDH | High | 24.4 | 47.82 | 1.96 | Lg. inc. | Lg. inc. | High | Rare | Good | Good | | | 1 | 22 |
| eastern hophornbeam; ironw | Ostrya virginiana | WSL | Low | 40.2 | 44.99 | 1.12 | No change | Sm. inc. | High | Rare | Fair | Good | | | 1 | 23 |
| striped maple | Acer pensylvanicum | NSL | Medium | 26.8 | 36.8 | 1.37 | Lg. dec. | Lg. dec. | Medium | Rare | Poor | Poor | | | 1 | 24 |
| blackgum | Nyssa sylvatica | WDL | Medium | 23.2 | 36.71 | 1.58 | Lg. inc. | Lg. inc. | High | Rare | Good | Good | | | 1 | 25 |
| pignut hickory | Carya glabra | WDL | Medium | 18.3 | 35.78 | 1.96 | Lg. inc. | Lg. inc. | Medium | Rare | Good | Good | | | 1 | 26 |
| sassafras | Sassafras albidum | WSL | Low | 12.2 | 31.96 | 2.62 | Lg. inc. | Lg. inc. | Medium | Rare | Good | Good | | | 1 | 27 |
| white spruce | Picea glauca | NSL | Medium | 6.1 | 31.44 | 5.16 | Very Lg. dec. | Very Lg. dec. | Medium | Rare | Lost | Lost | | | 0 | 28 |
| scarlet oak | Quercus coccinea | WDL | Medium | 12.2 | 28.83 | 2.36 | Lg. inc. | Lg. inc. | Medium | Rare | Good | Good | Infill ++ | Infill ++ | 1 | 29 |
| Norway spruce | Picea abies | NSH | FIA | 3.7 | 27.41 | 7.49 | Unknown | Unknown | NA | Rare | NNIS | NNIS | | | 0 | 30 |
| black locust | Robinia pseudoacacia | NDH | Low | 2.4 | 20.85 | 8.55 | Lg. inc. | Lg. inc. | Medium | Rare | Good | Good | | | 1 | 31 |
| red pine | Pinus resinosa | NSH | Medium | 3.7 | 19.99 | 5.46 | Lg. dec. | Very Lg. dec. | Low | Rare | Poor | Lost | | | 0 | 32 |
| green ash | Fraxinus pennsylvanica | WSH | Low | 3.7 | 18.33 | 5.01 | Sm. dec. | No change | Medium | Rare | Poor | Fair | | Infill + | 2 | 33 |
| Scotch pine | Pinus sylvestris | NSH | FIA | 4.9 | 15.06 | 3.09 | Unknown | Unknown | NA | Rare | NNIS | NNIS | | | 0 | 34 |
| American elm | Ulmus americana | WDH | Medium | 7.3 | 12.28 | 1.68 | No change | No change | Medium | Rare | Fair | Fair | Infill + | Infill + | 1 | 35 |
| bitternut hickory | Carya cordiformis | WSL | Low | 2.4 | 6.46 | 2.65 | Sm. dec. | Sm. inc. | High | Rare | Poor | Good | | Infill ++ | 2 | 36 |
| red spruce | Picea rubens | NDH | High | 1.2 | 5.65 | 4.63 | Lg. dec. | Very Lg. dec. | Low | Rare | Poor | Lost | | | 0 | 37 |
| black ash | Fraxinus nigra | WSH | Medium | 2.4 | 5.43 | 2.23 | Very Lg. dec. | Very Lg. dec. | Low | Rare | Lost | Lost | | | 0 | 38 |
| American chestnut | Castanea dentata | NSLX | FIA | 3.7 | 2.44 | 0.67 | Unknown | Unknown | Medium | Rare | FIA Only | FIA Only | | | 0 | 39 |
| mockernut hickory | Carya alba | WDL | Medium | 3.7 | 2.32 | 0.63 | Lg. inc. | Lg. inc. | High | Rare | Good | Good | Infill ++ | Infill ++ | 2 | 40 |
| slippery elm | Ulmus rubra | WSL | Low | 1.2 | 0.7 | 0.57 | Sm. inc. | Lg. inc. | Medium | Rare | Good | Good | | | 2 | 41 |
| shingle oak | Quercus imbricaria | NDH | Medium | 1.2 | 0.46 | 0.38 | Lg. dec. | Lg. dec. | Medium | Rare | Poor | Poor | | | 0 | 42 |
| mountain maple | Acer spicatum | NSL | Low | 1.2 | 0.22 | 0.18 | Lg. dec. | Lg. dec. | High | Rare | Poor | Poor | | | 0 | 43 |
| eastern redcedar | Juniperus virginiana | WDH | Medium | 0 | 0 | 0 | New Habitat | New Habitat | Medium | Absent | New Habitat | New Habitat | | Migrate ++ | 3 | 44 |
| shortleaf pine | Pinus echinata | WDH | High | 0 | 0 | 0 | New Habitat | New Habitat | Medium | Absent | New Habitat | New Habitat | | Migrate ++ | 3 | 45 |
| Virginia pine | Pinus virginiana | NDH | High | 0 | 0 | 0 | New Habitat | New Habitat | Medium | Absent | New Habitat | New Habitat | | Migrate + | 3 | 46 |
| cittamwood/gum bumelia | Sideroxylon lanuginosum ssp | NSL | Low | 0 | 0 | 0 | Unknown | Unknown | High | Absent | Unknown | Unknown | | | 0 | 47 |
| black hickory | Carya texana | NDL | High | 0 | 0 | 0 | New Habitat | New Habitat | Medium | Absent | New Habitat | New Habitat | | | 0 | 48 |
| sugarberry | Celtis laevigata | NDH | Medium | 0 | 0 | 0 | Unknown | New Habitat | Medium | Absent | Unknown | New Habitat | | | 0 | 49 |
| eastern redbud | Cercis canadensis | NSL | Low | 0 | 0 | 0 | New Habitat | New Habitat | Medium | Absent | New Habitat | New Habitat | | Migrate + | 3 | 50 |
| flowering dogwood | Cornus florida | WDL | Medium | 0 | 0 | 0 | New Habitat | New Habitat | Medium | Absent | New Habitat | New Habitat | | Migrate + | 3 | 51 |
| American holly | Ilex opaca | NSL | Medium | 0 | 0 | 0 | Unknown | Unknown | Medium | Modeled | Unknown | Unknown | | | 0 | 52 |
| black walnut | Juglans nigra | WDH | Low | 0 | 0 | 0 | New Habitat | New Habitat | Medium | Absent | New Habitat | New Habitat | Likely + | | 3 | 53 |
| sweetgum | Liquidambar styraciflua | WDH | High | 0 | 0 | 0 | New Habitat | New Habitat | Medium | Absent | New Habitat | New Habitat | | Migrate ++ | 3 | 54 |
| sourwood | Oxydendrum arboreum | NDL | High | 0 | 0 | 0 | New Habitat | New Habitat | High | Absent | New Habitat | New Habitat | Migrate + | Migrate + | 3 | 55 |
| sycamore | Platanus occidentalis | NSL | Low | 0 | 0 | 0 | New Habitat | New Habitat | Medium | Absent | New Habitat | New Habitat | Migrate ++ | Migrate ++ | 3 | 56 |
| cherrybark oak; swamp red o | Quercus pagoda | NSL | Medium | 0 | 0 | 0 | Unknown | Unknown | Medium | Absent | Unknown | Unknown | | | 0 | 57 |
| blackjack oak | Quercus marilandica | NSL | Medium | 0 | 0 | 0 | New Habitat | New Habitat | High | Absent | New Habitat | New Habitat | | | 3 | 58 |
| post oak | Quercus stellata | WDH | High | 0 | 0 | 0 | New Habitat | New Habitat | High | Absent | New Habitat | New Habitat | | Migrate ++ | 3 | 59 |
| winged elm | Ulmus alata | WDL | Medium | 0 | 0 | 0 | New Habitat | New Habitat | Medium | Absent | New Habitat | New Habitat | | | 3 | 60 |

**Figure 4.** HQ and CL output table for the Allegheny National Forest, Pennsylvania, sorted in decreasing order of current species abundance (FIAsum). Range indicates whether species is wide or narrow (W/N) in distribution, dense or sparse (D/S) in frequency of presence within FIA plots, and high or low (H/L) in importance value when it is found. MR refers to model reliability (see [21] for explanation). %Cell refers to the percentage of DISTRIB-II cells with the species present. FIAiv is the average importance value for the species when present on FIA plots. ChngCl45 or 85 presents the change classes (increase, decrease, or no change) of habitat suitability by 2100, according to RCP 4.5 (low emissions) or 8.5 (high emissions). Adapt is a class of adaptability of the species according to the ModFacs. Abund is an abundance class based on FIAsum. Capabil45 or 85 is the capability of the species to cope with the climates of RCP 4.5 or 8.5 at 2100, based on abundance, change classes, and adaptability. SHIFT45 and 85 show two classes of Infill (+ or ++) to indicate these species are currently found rarely in the NF and likely to expand in the next 100 years, while Migrate (+ or ++) indicate that the species did not occur on FIA plots, but SHIFT (RCP 4.5 or 8.5) did indicate potential for colonization in the NF within 100 years. We also show two classes of Likely (+ or ++) to show, because of signals within SHIFT outputs, that the species is likely present in the unit even though it was not found on the FIA plots within the unit. SSO is species selection option to assist in decisions regarding promoting the species, where 1 indicates the species is currently present and has at least a fair capability to cope, 2 indicates the species is rare or close to the NF boundary and has a good chance of spreading into the NF, 3 indicates the species is not recorded in FIA plots but does have some chance of getting colonized within 100 years, and 0 indicates further evaluation may be required. Finally, the N column simply is a counter. The table is explained fully in Supplementary File 3—Explanation of species tables.

The FIA plot data were tabulated within each unit (either $1 \times 1$ or NF) to yield a ranked list of tree species, by importance value (IV) quantified equally between total basal area and number of stems. The actual IVs, under **FIAsum** (bolded elements are headers for the output tables for each unit, example shown in Figure 4; see also Supplementary File 3—Explanation of species tables), were summed for each $10 \times 10$ or $20 \times 20$ km cell within the unit and recalibrated to the standardized area of 10,000 km$^2$ (the approximate area of a $1 \times 1$ degree grid at 36° N latitude), so that values are normalized and comparisons among units can be made on species importance; they then were assigned to one of four abundance classes (noted as **Abund**) according to these breakpoints: abundant (FIAsum >75), common (FIAsum 5–75), rare (FIAsum >0–5), and absent (FIAsum = 0). The IVs were also averaged for only those cells (noted as **%Cell**) that contained the species to yield the importance of the species only where it is present, not summed over the entire unit (noted as **FIAiv**). An examination of the IV for each species across its range allowed us to assign a **Range** classification. The Range field provides a quick indication if the species of interest is narrow or widely dispersed across North America (N vs. W), with Narrow meaning the species is found across <10% of DISTRIB-II grid cells in the eastern US and Wide >10%. The species is also categorized as found commonly (Dense if ≥40% of FIA plots among grid cells with IV >0) or rarely (Sparse if <40% of FIA plots have the species) within its overall distribution among eastern US FIA records. Finally, the ecological importance or abundance for each species can be indicated by the average importance value among the plots where the species is present, with High if average IV ≥6.0 (the median of mean IV across all species) and Low for values <6.0 [21]. We also assigned three levels of model reliability (**MR**) [21]. To do so, we evaluated and combined five model performance variables into a single rating: (1) a pseudo-R$^2$ obtained from the RandomForest (RF) model; (2) a Fuzzy Kappa comparing the imputed RF map to the FIA-derived map [22]; (3) a tree skill statistic of the imputed RF, after removing records with very high coefficient of variables (CV); (4) the deviance of the CV among 30 regression trees via bagging [23,24]; and (5) the stability of the top five variables from 30 regression trees [25].

By ratioing suitable habitat (IV) in future to suitable habitat at present, we can generate five classes of change (noted as **ChngCl45** or **ChngCl85**, depending on representative concentration pathways (RCP) 4.5 and RCP 8.5; Figure 3, [21]) according to the following break points in future:actual IV ratio: large decrease (<0.5), small decrease (0.5–0.8), no change (0.8–1.2), small increase (1.2–2.0), large increase (>2.0). For rare species, those occupying <10% of the region, the following break points were used: large decrease (<0.2), small decrease (0.2–0.5), no change (0.5–4.0), small increase (4.0–8.0), and large increase (>8.0). Ecologically, the assumption is that if the species is projected have increased (or decreased) summed IVs in the future, conditions will be more (or less) favorable for the species and is categorically represented by the change classes. The ModFacs assessment, based on 12 disturbance and 9 biological traits, provided a baseline rating as to the species' adaptability (**Adap**) to the changing climate (Figure 3, [19]). The disturbance factors address, based on literature surveys, how well the species is expected to cope with additional stresses from drought, flood, invasives, wind, fire, and the like into the future, while the biological factors address the capability of the species to regenerate vegetatively or via seed, its shade tolerance, its edaphic and habitat specificity, and the like. Adaptability scores were generated via weighted summations of climate-related biological and disturbance scores and classified into low, medium, and high adaptability [19] (see also Supplementary File 1—Climate Change Atlas). Notably, adaptability scores are based on the species' attributes across their entire range, but they may vary in their response to disturbances in certain locations; managers may adjust individual species scores according to local knowledge. The SHIFT model, which calculates colonization likelihood (CL) [11,26], was calibrated to approximate 50 km migration per century, an optimistic assumption across species [27]. Though migration is dependent on the dispersal characteristics of the species, and a single migration rate of 50 km/100 years is a broad assumption, insufficient information exists to defensively assign rates to individual species. We therefore use a historically defensible rate based on Holocene estimates [27,28].

## 2.3. Rules for Capability to Cope with a Changing Climate

Besides habitat suitability (ChngCl45 or ChgnCL85), additional information is desired to assess the species at a more specific spatial unit—the 1 × 1 or NF. To do so, we developed a rating scheme to include suitability, adaptability, and abundance to derive a capability for each species to withstand the challenges (and expand with opportunities) posed by the changing climate. We calculate a five-class capability rating (**Capabil45** or **Capabil85**) based on Abund, Adap, and ChngCl45 or ChngCl85 (Table 1). Following the capability scoring presented in Table 1, if Abund was 'Abundant', we enhanced the capability by one class (e.g., a rating of 'Good', became 'Very Good' in the final class). Similarly, if Abund was 'Rare', we decreased the rating by one class, and if Abund was 'Common', no alteration occurred. We somewhat arbitrarily promote or demote the capability one class based on abundance. We assume that within the large area (roughly 8000–10,000 km²) of landscape within each area of interest, there will be spatial variability of climate driven by topographic or geomorphic diversity [29] which will increase the probability of persistence under a changed climate. If the species is abundant now, we assume there is a greater potential for specific habitats to be suitable and/or genetic variants to be more resistant. We realize we do not have direct data to support these assumptions although landscape diversity has been linked to biodiversity and resilience [30]; we can only logically infer the assumption will be true for at least a portion of the species we model.

**Table 1.** Initial capability class ratings defined as colored classes, depending on the change class and adaptability of the species, and other classes for selected species. For final capability rating (Very Good to Very Poor): if abundance was 'Abundant', move up one class; if 'Rare' move down one class; if 'Common' stay in class.

| Change Class | Adaptability | | |
|:---:|:---:|:---:|:---:|
| | *High* | *Medium* | *Low* |
| **Large increase** | Very Good | Very Good | Good |
| **Small increase** | Very Good | Good | Fair |
| **No change** | Good | Fair | Poor |
| **Small decrease** | Fair | Poor | Poor |
| **Large decrease** | Fair | Poor | Very Poor |
| **Other capability classes:** | | | |
| | Lost | All suitable habitat lost | |
| | New Habitat | New habitat appearing | |
| | FIA Only | Unacceptable model for future, only FIA reported | |
| | NNIS | Non-native invasive species, only FIA reported | |
| | Unknown | Modeled as present, unknown | |

## 2.4. Rules for Expand, Migrate, and Likely, and Species Selection Options

We used CL in conjunction with HQ to generate HQ-CL classes of species for consideration for planting or other forms of promotion within the areal units of analyses (Figure 3). These are included in the fields **SHIFT45** (RCP 4.5) or **SHIFT85** (RCP 8.5) in the table outputs. 'Infill' represents those species that are currently rare in the unit but have high potential to expand within its boundaries. Two classes (Infill+ and Infill++) were generated based on the capability and the percent occupancy currently, plus those areas with at least a 50% CL within 100 years (Table 2). The 'Likely' classes indicate that the species was not found in FIA plots, but that there is a relatively high probability that the species exists in the unit, because either the species has some area with at least 50% CL (Likely+) or that it has both HQ-CL and at least 2% of the area has at least 50% CL (Likely++). The 'Migrate' classes indicate that the species was not found by FIA, but that the DISTRIB-II model indicates new habitat appears in the unit, with either low HQ and CL (HQ-CL = 1, Migrate+) or higher levels of these two components (HQ-CL > 1, Migrate++). Admittedly, this is a very liberal assessment for 'Migrate', as the area qualifies if only a fraction of the area of interest has only a 2% probability of colonization within 100 years.

Species selection options (**SSO**) provides options to assist in decisions regarding promoting the species, where 1 indicates the species is currently present and has at least a fair capability to cope with

the changing climate—thus suited for planting; 2 indicates the species is rare or in close proximity (SHIFT shows potential to reach the area within 100 years) to the 1 × 1 grid or NF boundary and has a good chance of spreading into the area—thus also suited for planting but perhaps not as suited as #1; 3 indicates the species is not recorded in FIA plots but does have some chance of getting colonized within 100 years—thus could be planted similar to 'Migrate+ or Migrate++' mentioned above; and 0 indicates none of the above but further evaluation may be required before eliminating consideration of the species for planting.

**Table 2.** Criteria for classes of Infill, Likely, and Migrate, according to the HQ and CL outputs. Capability and HQ-CL are determined separately for RCP 4.5 and RCP 8.5. %OccCol pertains to the percentage of the unit that is either already occupied or with at least a 50% probability of becoming potentially occupied (according to SHIFT) within 100 years. %2Col pertains to the percentage of the unit not already occupied that has at least a 2% probability of getting colonized within 100 years.

| Class | Capability | HQCL | %OccCol | %2Col |
|---|---|---|---|---|
| Infill+ | Fair, Poor | ≥1 | >0–50 | any |
| Infill++ | Good, Very Good | ≥1 | >10–50 | any |
| Likely+ | New Habitat, Unknown | any | >0 | any |
| Likely++ | New Habitat, Unknown | ≥1 | >2 | any |
| Migrate+ | New Habitat | =1 | 0 | >0 |
| Migrate++ | New Habitat | >1 | 0 | >0 |

*2.5. Mapping Aggregated 1 × 1 Outputs*

The 1 × 1 degree units represent approximately 10,000 km$^2$ at 36° latitude. Because of the Earth's spherical shape, the east–west width (and area) is smaller north of that line and larger south of that line. For smaller units of analysis (e.g., national forests or grasslands represented here), the unit was buffered so that a minimum of 8000 km$^2$ was represented for analysis (Figure 1). Thus, whenever we refer to a NF, we refer to this buffered (if necessary to reach 8000 km$^2$ minimum size) area. This minimum size was required so that an adequate number of FIA plots could be represented for each forest within the modeling grid [21]. The 1 × 1 degree units also represent ~100 10 × 10 km cells or ~25 20 × 20 km cells, or a mixture; the buffered NFs represent at least 80 10 × 10 km cells or 20 20 × 20 km cells—these numbers of cells represent sufficient samples for meaningful summing or averaging across the unit, and are the reason for choosing such a coarse analysis. As mentioned above, the FIAsum values were all standardized to equate to the 10,000 km$^2$ to enable cross-table comparisons of species importance; comparisons could therefore be made among the 469 1 × 1 units or the 55 NF units.

By combining information across the 469 1 × 1 tables, mapping of particular characteristics could be accomplished. To do so, counts were made of the number of species matching the criterion being queried for each 1 × 1 degree unit and aggregated across units to populate the maps. Though any of the tabulated variables described above could be mapped, we present below: (1) the number of total species quantified in each unit; (2) the number of oak (*Quercus* spp.) species, now and potentially in the future; (3) the number of species with at least some chance of migrating naturally into each unit under RCP 8.5, and thus higher candidates for artificial migration (Migrate+ plus Migrate++); and (4) the number of species that can be considered likely to expand under RCP 8.5, and thus higher candidates for infill planting (Infill+ plus Infill++).

**3. Results**

*3.1. Mapping Habitat Quality and Colonization Likelihood Outputs*

The combination of HQ and CL is exemplified for eastern hemlock (*Tsuga canadensis*) in Figure 5. First, the current condition is established using FIA data within the modeling grid (Figure 5a). Then potential suitable habitat (habitat quality, or HQ) by 2100 was mapped under RCP 8.5 (Figure 5b). The colonization likelihood (CL) map was independently created (Figure 5c), then intersected with the

HQ map to yield a map depicting those areas that have at least some chance of colonization within 100 years (Figure 5d). Both the HQ and CL maps were reclassified into three possible classes (low, medium, high), with IV breakpoints of HQ at 5 (break between low and medium) and 15 (between medium and high) and CL breakpoints at 10 and 50 percent. Merging the two outputs can produce up to nine combinations; for eastern hemlock, there was no high HQ so only six HQ-CL combinations are shown (Figure 5d). Finally, the combined HQ-CL information was used to identify those areas where the species could 'infill', or fill in cells that have no FIA evidence that it exists now, but is closely surrounded by the species (Figure 5, Infill inset map). Or, if the species could have some potential to move into new habitat not previously occupied (most often to the north of the current boundary), those areas can be noted as 'migrate' areas (Figure 5, Migrate inset map). We also can identify those species that are 'likely' present but missed by FIA (these locations would appear within the 'infill' zones on the map but cannot be precisely located). Thus, the combination of HQ and CL results not only identifies potential changes in suitable habitat under various scenarios of climate change, but also provides, for each species present currently or potentially in the future, estimates of CL from natural migration within 100 years.

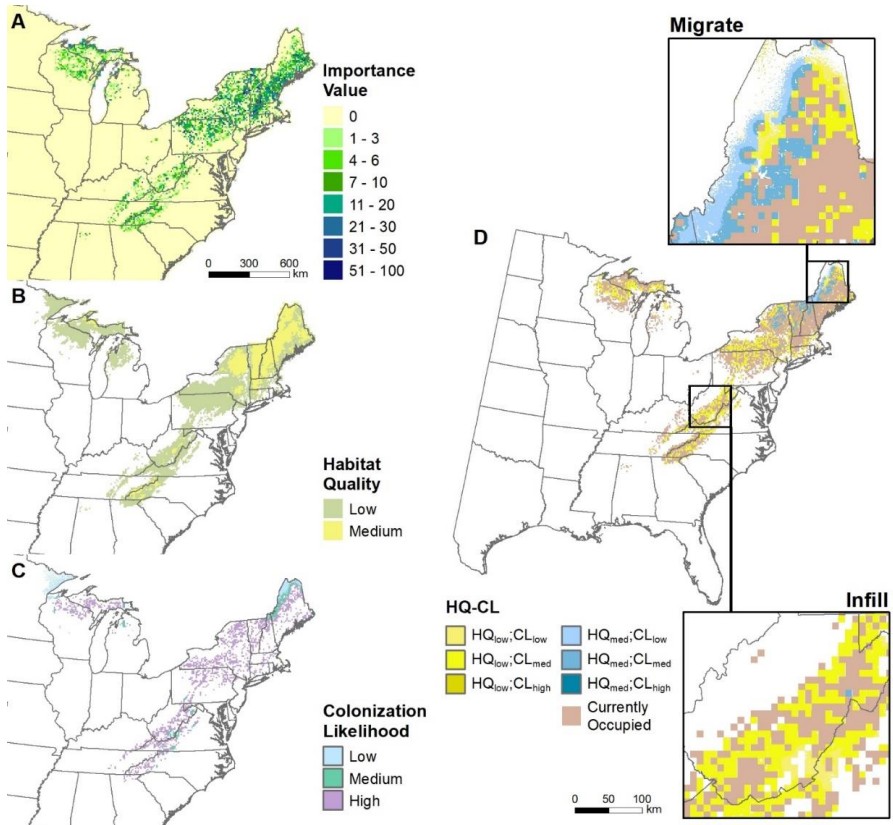

**Figure 5.** Process flow to intersect HQ and CL for eastern hemlock. (**a**) Current importance value of hemlock as determined by FIA data; (**b**) habitat quality (HQ) at RCP 8.5, year 2100, reclassed into low, medium, high HQ (there was no high HQ for hemlock), at a resolution of $10 \times 10$ or $20 \times 20$ km; (**c**) colonization likelihood (CL) into RCP 8.5 habitat, and reclassed into low, medium, high CL, at a resolution of $1 \times 1$ km; (**d**) combination of (**a–c**), yielding locations with null to high CL on top of low to medium HQ, as well as currently occupied cells; (infill inset) detail of locations where infilling is primary; (migrate inset) detail of locations where migrating is primary.

## 3.2. Species Summaries by National Forest

We present our primary tabular output for one NF, the Allegheny NF (coded '1' in Figure 1 and record 1 in Table 3), but similar tables have been prepared for each $1 \times 1$ degree area and are available

for download. Other units of analysis will eventually be available including national parks, watersheds, ecoregions, states, and others. Our combination of HQ, CL, Adap, and current FIA estimates of importance value allow a detailed presentation of (1) species importance currently; (2) the potential changes in suitable habitat by 2100; (3) the adaptability of each species to the changing climate; (4) the capability of each species to cope with the 2100 climate based on adaptability and abundance currently within the NF; (5) the likelihood of each species to naturally migrate into the NF; and (6) an assessment of the potential for the species to be used for planting or otherwise promoting within the NF. These are all presented within an information-packed, but easily unpacked table (Figure 4).

**Table 3.** Identity code (noted in Figure 1), coordinates, actual area, and buffered area (km$^2$) of each national forest or grassland. National grasslands are noted as 'NG'; all others are national forests. Also shown are results for number of species recorded by FIA within the buffer region, and the number of species potentially infilling, likely, and migrating according to the RCP 8.5 scenario of climate change (see text for explanation). For simplicity, only RCP 8.5 data are shown.

| Identifier | Name | Longitude | Latitude | Area, km$^2$ | Buffer Area, km$^2$ | Number Species | Number Infill | Number Migrate |
|---|---|---|---|---|---|---|---|---|
| 1 | Allegheny | −79 | 41.7 | 2996.5 | 8200 | 43 | 6 | 11 |
| 2 | Angelina | −94.4 | 31.3 | 1602.8 | 8800 | 61 | 22 | 1 |
| 3 | Apalachicola | −84.7 | 30.2 | 2555.9 | 8044 | 64 | 18 | 1 |
| 4 | Bienville | −89.5 | 32.3 | 1572 | 8900 | 67 | 16 | 2 |
| 5 | Black_Kettle NG | −99.5 | 35.7 | 982 | 8196 | 18 | 2 | 6 |
| 6 | Caddo NG | −96 | 33.6 | 277.3 | 9100 | 46 | 13 | 0 |
| 7 | Chattahoochee | −84.2 | 34.7 | 5950.2 | 9700 | 72 | 24 | 9 |
| 8 | Chequamegon | −90.8 | 46 | 4209.7 | 11,815 | 40 | 5 | 9 |
| 9 | Cherokee | −83.1 | 35.8 | 5950.2 | 9600 | 74 | 35 | 9 |
| 10 | Chippewa | −94.1 | 47.4 | 6465.5 | 8800 | 34 | 6 | 11 |
| 11 | Conecuh | −86.6 | 31.1 | 692.9 | 8500 | 60 | 18 | 4 |
| 12 | Croatan | −77.1 | 35 | 1243.7 | 8034 | 62 | 17 | 2 |
| 13 | Daniel_Boone | −83.9 | 37.3 | 8282.7 | 14,100 | 77 | 10 | 4 |
| 14 | Davy_Crockett | −95.1 | 31.3 | 1602.8 | 8400 | 55 | 15 | 1 |
| 15 | Delta | −90.8 | 32.8 | 490.7 | 8000 | 62 | 30 | 3 |
| 16 | DeSoto | −89 | 31.1 | 3252.4 | 8700 | 67 | 18 | 0 |
| 17 | Finger_Lakes | −76.8 | 42.6 | 58.4 | 8300 | 62 | 11 | 8 |
| 18 | Francis_Marion | −79.8 | 33.3 | 1696.1 | 8221 | 70 | 24 | 1 |
| 19 | George_Washington | −79.3 | 38.2 | 7270.9 | 14,300 | 73 | 16 | 10 |
| 20 | Green_Mountain | −73 | 43.4 | 2543 | 8800 | 42 | 6 | 14 |
| 21 | Hiawatha | −86 | 46.2 | 5197.3 | 9549 | 35 | 7 | 14 |
| 22 | Holly_Springs | −89.2 | 34.7 | 1906 | 8500 | 70 | 13 | 4 |
| 23 | Homochitto | −90.9 | 31.5 | 1540.6 | 8400 | 72 | 17 | 2 |
| 24 | Hoosier | −86.5 | 38.5 | 2612 | 8100 | 65 | 11 | 4 |
| 25 | Huron | −84 | 44.6 | 2787.3 | 8078 | 39 | 6 | 12 |
| 26 | Jefferson | −81.2 | 37.1 | 4926.7 | 15,100 | 76 | 11 | 8 |
| 27 | Kisatchie | −92.8 | 31.7 | 4195.7 | 9800 | 74 | 12 | 2 |
| 28 | Lyndon_B_Johnson NG | −97.6 | 33.4 | 465.5 | 9200 | 32 | 14 | 1 |
| 29 | Manistee | −85.9 | 43.9 | 5377.5 | 8566 | 47 | 7 | 6 |
| 30 | Mark_Twain | −91.7 | 37.3 | 12,237.4 | 23,900 | 66 | 15 | 5 |
| 31 | Midewin NG | −88.1 | 41.4 | 104.3 | 8350 | 31 | 19 | 12 |
| 32 | Monongahela | −79.9 | 38.6 | 7270.9 | 10,500 | 61 | 8 | 11 |
| 33 | Nantahala | −83.6 | 35.2 | 5950.2 | 9700 | 65 | 17 | 9 |
| 34 | Nicolet | −88.7 | 45.7 | 3890.7 | 8200 | 39 | 8 | 6 |
| 35 | Ocala | −81.8 | 29.2 | 1791.6 | 8177 | 42 | 4 | 5 |
| 36 | Oconee | −83.5 | 33.4 | 1049.8 | 8700 | 62 | 10 | 3 |
| 37 | Osceola | −82.5 | 30.4 | 932.4 | 8400 | 44 | 12 | 6 |
| 38 | Ottawa | −89.2 | 46.5 | 3890.7 | 9559 | 36 | 7 | 11 |
| 39 | Ouachita | −93.9 | 34.6 | 9682.3 | 15,000 | 62 | 16 | 11 |
| 40 | Ozark | −93.5 | 35.7 | 9682.3 | 11,800 | 70 | 13 | 11 |
| 41 | Pisgah | −82.4 | 35.8 | 5950.2 | 10,100 | 70 | 12 | 12 |
| 42 | Sabine | −93.9 | 31.5 | 1602.8 | 8500 | 63 | 22 | 3 |
| 43 | Saint_Francis | −90.7 | 34.7 | 120.5 | 8700 | 63 | 45 | 3 |
| 44 | Sam_Houston | −95.4 | 30.5 | 2005.3 | 8200 | 63 | 18 | 1 |
| 45 | Shawnee | −88.9 | 37.5 | 3501 | 8000 | 73 | 13 | 5 |
| 46 | Sheyenne NG | −97.2 | 46.4 | 551.6 | 8400 | 9 | 6 | 7 |
| 47 | Sumter | −82.1 | 34.3 | 5950.2 | 8200 | 72 | 24 | 1 |
| 48 | Superior | −91.6 | 47.8 | 13,204.5 | 16,415 | 32 | 11 | 14 |
| 49 | Talladega | −86.5 | 33.2 | 3042.6 | 9700 | 68 | 15 | 7 |
| 50 | Tombigbee | −89.2 | 33.7 | 713.1 | 8700 | 69 | 20 | 2 |
| 51 | Tuskegee | −85.6 | 32.5 | 63.2 | 8100 | 70 | 21 | 3 |
| 52 | Uwharrie | −79.9 | 35.4 | 889.2 | 8200 | 73 | 18 | 4 |
| 53 | Wayne | −82 | 39.3 | 3451.8 | 10,800 | 71 | 13 | 5 |
| 54 | White_Mountain | −71.4 | 44.2 | 3511.8 | 8800 | 37 | 9 | 14 |
| 55 | WB_Bankhead | −87.3 | 34.2 | 1409.3 | 8900 | 71 | 0 | 0 |

### 3.3. Trends in Area and Species Counts for National Forests and Grasslands

Besides identifying the locations (labels for Figure 1) for the 55 national forest and grasslands, Table 3 presents the geographic coordinates and size information of both the NF itself and the buffer area used in evaluating the species for this effort. Ranging from 104.3 km$^2$ for the Midewin NG to 13,204.5 km$^2$ for the Superior NF, there is a 126-fold variation in area from smallest to largest. To dampen this large variation and provide sufficient area for multiple FIA plots, we added area to accumulate a minimum of 8000 km$^2$ surrounding each NF, so the range in buffered area was 8000 km$^2$ for the Delta NF to 23,900 km$^2$ for the Mark Twain NF (consisting of several units merged together in the buffering process). Though some species–area curve impacts could be present, a correlation between the buffered area and species count was only 0.14 (NS).

Tree species counts (according to FIA) for the various NFs ranged from 9 (for the Sheyenne NG in North Dakota) to 77 (Daniel Boone NF in Kentucky), and were negatively correlated to latitude ($r = -0.57$, $p < 0.001$); southern NFs generally had higher species richness (Table 3). Also recorded are the number of 'infill', 'likely', and 'migrate' species (either + or ++) under RCP 8.5. The table indicates that up to eight species were likely present in the NF but missed by FIA for the Green Mountain NF, up to 45 species (St. Francis NF) have potential to infill, or expand importance within that forest, and up to 14 species (Green Mountain NF) have potential to migrate into this buffered NF from points south (Table 3). The count of potential species to migrate was highly correlated with latitude ($r = 0.72$, $p < 0.001$), indicating that there is a greater potential for assisted migration of tree species into the more northerly NFs, should that be desired.

### 3.4. Questions and Answers for Example NF: Allegheny NF

To simplify interpretation of the table outputs, we have developed a series of 17 questions that can be asked of the data table. Users can peruse the questions and explanations to see what might align with their interests. The methods to (a) extract the answers and (b) provide answers for one unit, the Allegheny NF, are presented in Appendix A. From this example, many insights on the current and potential state of tree species on the Allegheny NF can be gleaned. A subset of the questions are as follows, which can be asked of any 1 × 1 degree or national forest area:

1. How many species are known (by FIA) to be present currently, what are these species, and how abundant are they?
2. By 2100, how might suitable habitat change for trees under low and high emission scenarios?
3. How adaptable are the tree species to conditions expected to change under climate change?
4. How capable might the species be with regards to coping with the changing climate, and what species may be most (or least) vulnerable?
5. Within the next 100 years, what species may be able to naturally migrate into the area?
6. What species, now rare, are suited to expand or infill in importance over the century?
7. What species are likely present in the area, though missed by FIA plots?
8. What species might be useful for selecting as candidates for planting in the face of a changing climate?

For the Allegheny example, we show that 43 species are currently present (6 abundant, 15 common, and 22 rare species), though one additional species is likely there, and an additional 14 species have new habitat appearing by end of century. Of these, 13 are likely to lose suitable habitat under RCP 8.5, while 17 gain habitat. Still, 25 species appear to have good or very good capability to cope with the changing climate, while the most vulnerable species appear to be red spruce, red pine, black ash, pin cherry, quaking aspen, and white spruce. Nine species show good potential to migrate into the area under the high emission scenario, while six additional species are marked as potentials to infill into surrounding locations from their currently relatively uncommon status. Using the species selection options, we found 30 species common and well suited for planting, four species uncommon but with

good adaptability, and 12 species potentially suited for planting, though with more risk of planting failure. Further details are provided in Appendix A and with the full table available as Supplementary File 2—Allegheny table.

By providing answers to the 17 questions for any national forest or 1 × 1 degree grid, a user is able to delve into many aspects of their forest attributes, both now and potentially into the future. Managers and the casual user alike will have species lists showing which are abundant, common, and rare within their area of interest. Managers will have data to support (or not) their decisions related to adaptation to the changing climate. For example, selecting species that may be candidates for planting may be better justified if the HQ-CL data suggest that the suitable habitats are ripe for the species in their area of interest. This is especially true for selecting species not now identified from their area of interest—picking species with 'Migrate+' or 'Migrate++' would seem more justified in that the habitat will likely be present in coming decades and the species is close enough now to provide some chance of natural migration into the area even without assistance.

### 3.5. Mapped Summaries for 1 × 1 Degree Grid

Tree species relative richness, as recorded on FIA plots and summed for each 1 × 1 degree grid, shows a remarkable variation in both latitude and longitude (Figure 6A). These patterns show maximum diversity in the southeastern portion of the U.S., with the maximum count of 82 species found on the southern Mississippi–Alabama border. Lowest species richness was found, as expected, in the prairie regions of North and South Dakota, where very few FIA plots were placed in forested (mostly riparian) zones.

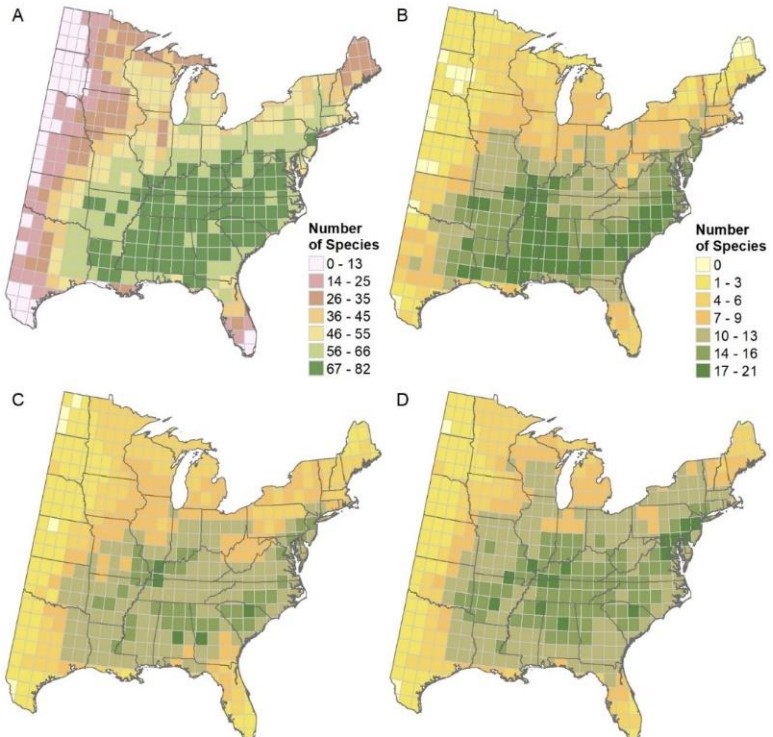

**Figure 6.** (**A**) The number of species recorded by FIA by 1 × 1 degree cell across the eastern US. (**B**) The number of oak (*Quercus* spp.) species recorded by FIA by 1 × 1 degree cell across the eastern US. (**C**) The number of oak (*Quercus* spp.) species projected to have suitable habitat (HQ) and at least some chance of colonization (CL) at 2100 according to RCP 4.5. (**D**) The number of oak (*Quercus* spp.) species projected to have suitable habitat (HQ) and at least some chance of colonization (CL) at 2100 according to RCP 8.5. Figures (**C,D**) use same figure legend as (**B**).

Diving further into genus-level analyses, the oaks (*Quercus* spp.) show highest richness in the far south central portion of the country, maxing out at 21 species; lowest oak richness was in both the northwestern and the northeastern portions of the study area, as well as southern Florida (Figure 6B). To demonstrate potential changes under climate change, we also present projected counts of the number of oaks with suitable habitat and colonization potential under RCP 4.5 (Figure 6C) and RCP 8.5 (Figure 6D). In comparison with the current oak distribution (Figure 6B), the future (~2100) models show potentials for subtle decreases in oak species counts in the south and slight increases towards the north. This trend is more prominent with RCP 8.5 (Figure 6D).

The number of species with potential to migrate, by 1 × 1 cell and for RCP 8.5, is depicted in Figure 7A. There is a large north to south gradient in species numbers; the correlation with latitude is 0.72 ($p < 0.001$). The Plains States and Corn Belt states also have fewer species available for migration. The maximum number of potential species, according to this analysis, that may be appropriate to plant in an assisted migration mode, is 19, from the Adirondack Region of New York, with many other cells in the far north also with high numbers of species with potential to migrate in (Figure 7A).

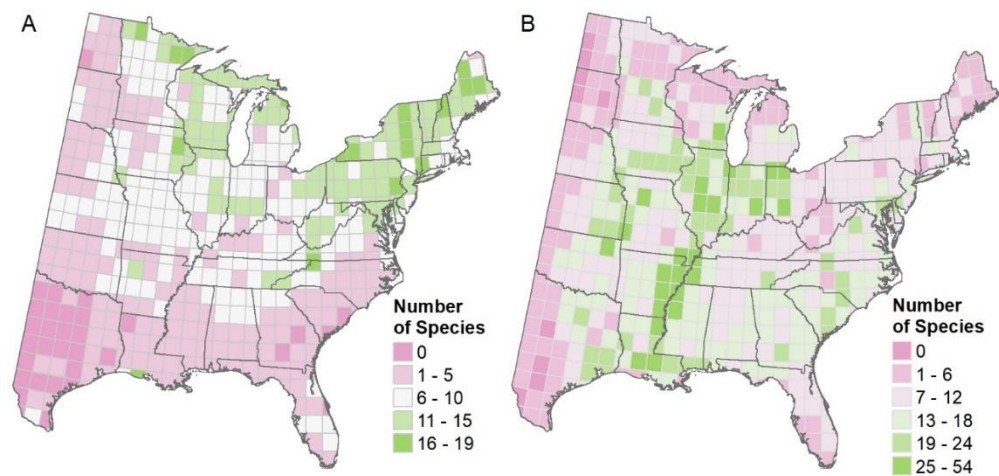

**Figure 7.** (**A**) The number of species that may potentially migrate into each 1 × 1 degree cell (from elsewhere), within 100 years across the eastern US (Migrate+ plus Migrate++), under RCP 8.5. (**B**) The number of species with potential to infill (Infill+ plus Infill++) into each 1 × 1 degree cell, within 100 years, across the eastern US.

The number and distribution of species with potential to infill provides a major contrast to the migration map, in that numbers can range up to 54 species, and with the highest numbers in the highly agricultural zones (Figure 7B). The maximum 1 × 1 values appear in the southern Mississippi floodplain, in highly agricultural zones (with few forested FIA plots so only a small total area was sampled) near locations with relatively high species richness. The correlation between Infill85 and Migrate85 was −0.34 ($p < 0.01$), further indicating the inverse relationship between the two, i.e., if the unit is highly forested, there are less species 'available' for infilling as they have already been captured by the FIA sampling.

Many other maps depicting trends among the 1 × 1 degree grids have been generated, and these reveal patterns associated with the numbers of abundant or rare species, the numbers of species within several common genera, the numbers of species most capable to deal with the changing climate, the top three species within each cell, and information related to the range distribution, among others. These maps and others will be made available for users to spatially compare among 1 × 1 degree grids relative to their areas of interest.

## 4. Discussion

This work brings together two lines of effort that our group has pursued for over two decades: that of the projected changes in habitat suitability under the changing climate [19,20,23,25,31–40] and that of the potential colonization likelihood into suitable habitat over the course of a century [11,26,41–43]. Advances in each line were achieved, and often necessitated, by enhancements or updates in data, climate projections, software, and computing capabilities. By combining these lines of effort for most of the tree species in the eastern US, we provide a mechanism to evaluate, with more reality imbedded, the potential outcomes at end of century. The intention of this work is to assist managers, both at the scale of regions or the entire eastern US as well as individual national forests or locations within 1 × 1 degree units, in adapting to a changing climate. We hope that the information outlined here can be used to assist in the silvicultural treatments available to managers, such as plantation, assisted migration, thinning, or natural regeneration, to help forests mitigate and adapt to the impacts of climate change [44]. This is the focus of the Northern Institute of Applied Climate Science, and the Climate Change Response Framework (www.forestadaptation.org) [45]. As laid out by Millar and Stephenson [46], adaptation pathways lead to resistance, resilience, or response (or facilitation, or transition). A national experiment is testing these approaches at various sites across the continent [47], and the data provided here can help in ascertaining species appropriate for each of these pathways, anywhere in the eastern United States. However, caution is advised when applying these results because no model can capture the full range of complexity acting on species through uncertain times, and although species suggested here as potentials for assisted migration may be adapted climatically to future climate conditions, other factors such as competition or herbivory may override other traits and restrain regeneration success [48] (see also Supplementary File 1—Climate Change Atlas). Additional caveats more specific to these results include the averaging over large land areas (>8000 km$^2$), the assumption of an average 100-year migration rate for all species of 50 km in uninterrupted forest, and the use of projected future:actual IVs as a proxy for future importance of the species. Thus the need exists for multiple, adaptive strategies, and interpretation and species selections by local experts, as we move into this changing, but uncertain future [49,50].

### 4.1. Applications for Summary Tables

This indices-based, tabular approach, while quite formidable upon first glance, is a comprehensive way to combine the outputs from 125 species models plus 23 non-modeled species (but with FIA information) into succinct, spatially explicit summaries for users of the data as a tabular substitute for the multiple individual maps. In this way, users can glean comprehensive information about the tree species currently present, or potentially present in the future, in their area of interest. Though our group and other groups have been modeling potential species changes of tree habitats for quite some time, this approach gives a condensed summary of species' current and potential future status for any area of interest. Beyond estimates of habitat suitability changes or range boundary extensions, this work adds reality in that natural species migrations will greatly lag the changing climatic conditions [42,51–54]. Estimates for end-of-century habitats and potential shifts provide information that managers can use now as they ready their forests for end-of-century climates.

The 1 × 1 summary tables are available for download or from the authors for any geographic location in the eastern US. The 469 1 × 1 degree files provide a wall-to-wall coverage and are named by their southeast boundary, such that, for example, a GPS coordinate of 42.44° N latitude, −82.55° W longitude will be named S42_E82. Refer also to Figure 1 for assistance in locating the appropriate table(s). Tables for the national forests and grasslands are also available. We intend to make tables available as well for national parks, watersheds, ecoregions, and other locations; the purpose is to provide a wealth of sortable information for managers, researchers, landowners, and interested publics.

The set of questions and answers for the Allegheny NF (see Appendix A) can be asked of any of the tables and represents some of the ways in which users may apply the data. Users interested in the current biodiversity status in their area of interest may glean the number, abundance, area occupied,

and identity of tree species that have been reported by FIA, or likely to be there even if not reported by FIA. The Range field provides a quick indication if the species of interest is narrow or widely dispersed, and how common or ecologically important it is within its range. Users interested in how species may generally react to the changing climate in their area of interest would focus more on the habitat suitability (ChgCl45 or ChgCl85) and modification factors (Adap).

The capability analysis, which incorporates habitat suitability, abundance, and adaptability, provides an assessment of the species' capability to withstand and cope under the expected conditions modeled for RCP 4.5 or 8.5. Thus, users can have a better idea of how the species may fare to 2100 in their particular area of interest.

Users interested in planting in the face of the changing climate also have resources available from the tables, in fields SHIFT45, SHIFT85, and SSO. Those users wanting to "hedge their bets" on planting species already reported by FIA in the area but quite uncommon (i.e., rare), can use the Infill markers to identify candidate species. Because of the hit or miss nature of sampling trees in highly agricultural regions like the Corn Belt or Mississippi floodplain, many of the Infill species listed for those areas will be quite common in the fencerows, woodlots, or riparian regions of the area, just missed by the few FIA plots landing on forests. Thus, an extra level of screening or targeted sampling may be required for these areas to account for the fewer plots, keeping in mind also that the model reliability varies among species represented on the tables. Of course, these highly agricultural zones will likely remain highly agricultural and the potential to Infill naturally is highly improbable into those regions. In these cases, managers could choose among the Infill species should they wish to afforest some of these agricultural areas.

The Migrate classification reveals those species that FIA did not report for the area, but the habitat will likely be suitable by 2100 and the species has some, often very limited, potential to migrate into the area within 100 years. The assumption is if the species could have any chance of migrating into the area naturally, it is likely a better candidate for assisted migration than if the species has potential future suitable habitat but the sources for natural migration are long distances away. Thus, we would encourage managers to look at these 'Migrate' species first if they wish to assist in transitioning their area for adaptation to climate change. Forestry assisted migration has long been practiced by silviculturalists [55], though it must be practiced with care [50]. The Infill and Migrate classes, along with the SSO classes, provide some guidance for species selection, but again, we emphasize that these lists are a starting place for decision making, in need of expert interrogation.

Further information regarding species selection options (SSO) is also provided via 0–3 scoring. Some species will be common already in the area of interest, and have qualities—of fair, good, or very good capability class—that provide the expectation that the species will also do fine even under the changed climate of 2100 (SSO = 1). An additional set of species may be present or near the area of interest currently, are usually quite rare, yet are in a position to potentially expand over time (SSO = 2). Because these species are 'rare', it is a similar, but a more restrictive set of species as compared to 'Infill' mentioned above as they can be 'common' species. A third set of species (SSO = 3) are those not present currently according to FIA ('new habitat' capability class) but may actually be there ('likely' species) or have a least a minimal potential to colonize over 100 years ('migrate' species). Users can tighten the selection criteria by using another variable available in the long file, %2Col, which only counts $1 \times 1$ km cells as colonized if the SHIFT model outputs at least 2% of the area of interest as colonized over 100 years (i.e., the cell gets colonized at least 2 times over the course of 100 runs from SHIFT). For example, one could consider only those species that have at least 5% of the area with at least a 2% probability of colonization (%2Col >5, see long definitions on spreadsheet, e.g., Supplementary File 2—Allegheny table).

We emphasize that these analyses are only to be used as general guidelines for species selection. The models are built from FIA data across the eastern US, and for certain species, local influences (e.g., lake effects) will override the general tendencies across the entire eastern US. The models also necessarily are built from coarse-level data and are unable to zero in on special or rare habitats, or may

have low model reliability (MR). Therefore, decision makers should use local knowledge to select species, even if they are not coded 1, 2, or 3 on the Species Selection Option (SSO), as they may still be suited for particular niches in a project area that helps meet overall objectives. Conversely, each species coded 1, 2 or 3 for SSO should be evaluated in the local context as they may not be suitable for particular sites.

Important also is consideration of the context of migration, and the role of migration, assisted or not, in overall biodiversity management under climate change. Species will be moving but not all will be perceived as valuable to the communities into which they establish. Though not so problematic for the 125 tree species presented here, an evaluation of each species for persecuting, protecting, or ignoring under the changing climate is warranted [56].

The spreadsheets available for the NFs and 1 × 1 degree grids have a total of eight worksheets for each unit which enable the user to better interpret the dense amount of information contained therein. These include the short-table and definitions (discussed above), questions to ask of the data, interpretation help, and some references. Also included is a long table with definitions, which includes all the short table fields but also a large number of fields that are available for users to delve into the actual numbers used to derive many of the categorical fields in the short table (note: this information is also presented in Supplementary File 3—Explanation of tables).

### 4.2. Mapped Summaries for 1 × 1 Degree Grids

The mapped summaries shown here, Figures 6 and 7, provide examples of the kind of outputs made possible by this wall-to-wall analysis. Any of the fields discussed above can be mapped to derive spatial patterning across the eastern US. Some patterns are expected, such as the species richness pattern (Figure 6A), which shows diminishing richness moving westward and northward. This pattern has been explored previously (e.g., [57,58]), with richness in concert with energy and moisture patterns.

Concomitant with overall richness is the pattern of oak species richness, a very diverse genus of up to 21 species per 1 × 1 degree grid in the south central states of Louisiana, Mississippi, and Alabama. Oaks are a foundational species in much of the eastern forests [59], and an extremely important genus for wildlife and botanical diversity [60,61] as well as a valuable timber resource [62]. For example, Tallamy and Shropshire [63] reported that more than 500 Lepidopteran species in the mid-Atlantic region use oaks. It is unfortunate that oaks, almost throughout their eastern North American range, are experiencing a regeneration problem whereby they are being replaced by more mesophytic species like maples (*Acer* spp.) that are more tolerant to shade but less tolerant to fire (that now rarely occurs) [64–66]. Though silvicultural treatments of repeated fire and partial harvest are tools that have been shown to promote oaks [67,68], these tools are difficult (i.e., expensive) to use across large swaths of land, especially when the majority of eastern forestlands are in private ownership. In contrast, models of climate change, such as those reported by our group since 1998 [20,25,36] and others [69–71] have consistently reported that oaks as a group should do well in a warmer and more drought-prone future. Nonetheless, it is imperative to conduct the silviculture to maintain a thriving oak component now so that propagules are available into the future [72] and enable the perpetuation of the rich oak diversity revealed in this 1 × 1 degree examination.

The number of species with potential to migrate reveals the dramatic but expected trend of more potential species for migrations north vs. south (Figure 7A). Obviously, there are fewer species to draw from locations further south in the southern latitudes; the Gulf of Mexico marks the end of terrestrial habitat. In contrast, the northern locations are somewhat cumulative with regards to possible sources of species to the south, in that the species-rich sections of the south also have less fragmented corridors for northward movement [73] and some species have potentials for long-distance dispersal (up to 500 km within the SHIFT algorithm). The highly fragmented forests within the Plains and Corn Belt of the Midwest also reduce the potential for species to migrate, i.e., the SHIFT model does not have many propagule sources when most of the land is under cultivation. Also noticeable when viewing the juxtaposition of the national forests (Figure 1) with the potential 'migrate' species of Figure 7A,

is the probable large influence of the NFs in providing possible sources of species for migration. Higher number of species with a potential to migrate are found in proximity to NFs in especially the southern Appalachians, the southern tier of NFs across Missouri, Indiana, Illinois, and Ohio, and the Northwoods of Minnesota, Wisconsin, and Michigan.

Species may also expand, or 'infill' in suitable locations within their current distribution. With up to 54 species depicted in $1 \times 1$ degree grids in some southern locations, and with very high numbers also in the highly agricultural zones (Figure 7B), there is a major contrast with the migration map. These values are high because forests are sparse and forested FIA plots are infrequent so that either (1) FIA missed the disparate locations that have the species; or (2) land-use conversion and fragmentation have eliminated the species from that unit. Managers can select from these lists for species potentially capable of dealing with the changing climate and are already in the vicinity albeit in low numbers.

Of course, any species identified via these tables must be vetted by local experts as to their suitability according to local ecological conditions. Because of the scale of analysis (8000 km$^2$ for national forests or 10,000 km$^2$ for $1 \times 1$ degree grids), there will always be large variations in soils, topography, land use, and hydrology within the unit that precludes or includes species as candidates. The online Climate Change Atlas website (www.fs.fed.us/atlas), as well as many other documents and web sites, also include ecological characteristics of species to help sort out the applicability of species for particular planting sites.

## 5. Conclusions

This study is a synthetic effort of over two decades of research and focuses on combinations of tree species occupying, currently or potentially in the future, each national forest and each $1 \times 1$ degree grid throughout the eastern US. We have combined 125 modeled species, both with regard to potential changes in suitable habitat and capability, but also with regard to the potential to migrate or infill naturally (according to historic migration rates) over the next 100 years. This resulting summarization effort provides an enormous data set, here described for 469 $1 \times 1$ degree grids and 55 national forests and grasslands (and we are compiling them for hydrologic units, ecoregions, states, and national parks as well). These data are available from authors and available for download at https://doi.org/10.2737/Climate-Change-Atlas-Combined-v4. We emphasize that these tables and maps are only the first-line estimate of potential species trends, intended to provide the managers with some tools to reduce the vast set of decisions before them as they proactively manage in the face of climate change.

**Supplementary Materials:** The following are available online at http://www.mdpi.com/1999-4907/10/11/989/s1, Supplementary File 1—Climate Change Atlas, Supplementary File 2—Allegheny table, Supplementary File 3—Explanation of tables.

**Author Contributions:** All four authors cooperated on conceptualization, methodology, validation, formal analysis, investigation, data curation, visualization, review and editing, and preparing proposals for funding. L.R.I. prepared the original draft, and supervised and administered the overall project beginning in 1995.

**Funding:** This research was funded by the USDA Forest Service, Northern Research Station, the Northern Institute of Applied Climate Science, and the USDA Climate Hubs as part of their appropriated funding.

**Acknowledgments:** The authors are indebted to the hundreds of FIA staff responsible for acquiring and processing data to make it available to researchers like ourselves. Funding for this project was provided by the USDA Forest Service Northern Research Station; this research did not receive any specific gran from funding agencies in the public, commercial, or not-for-profit sectors. Thanks to the reviewers of earlier drafts: Andrew Maday, Bryce Adams, Patricia Leopold, and Chris Swanston. Special thanks are due the reviewers of the original submitted manuscript; your comments were invaluable in strengthening the paper.

**Conflicts of Interest:** The authors declare no conflict of interest. The funders had no role in the design of the study; in the collection, analyses, or interpretation of data; in the writing of the manuscript, or in the decision to publish the results.

**Appendix A**

*Appendix A.1. Questions and Answers for Example NF: Allegheny NF*

An assessment of the output data for the Allegheny NF provides several observations both on current and potential future species. We present them in the form of answers to questions that may be asked of the data, then the way to extract the answer (a), and the answer for the Allegheny NF (b). These questions are all presented within the excel tables (on a sheet labeled 'Questions of tables') accessible with the online database. Though an extensive list of questions, it exemplifies the breadth of information possible to be gleaned for any areal unit via these tables; we have grouped the questions according to learning about the current status, the potential change in suitable habitat, and the potential change in migration status over 100 years.

Appendix A.1.1. Current Status of Species

1.  Of the species modeled, how many are known (by FIA) to be present in this NF?

    a.  The column **N** counts the species in this table. Some are present and some are modeled to have suitable habitat appear by 2100. Those with **%Cell** > 0 are known by FIA to be present. Some others may be present but were not found on FIA plots.
    b.  A total of 43 species identified by FIA for this NF.

2.  What species are present in this area (according to FIA)?

    a.  Look at **Common Name** or **Scientific Name**, those with **%Cell** >0 have FIA indication of presence.
    b.  Red maple, black cherry, sugar maple, and American beech top the list of 43 species.

3.  How common is each species, when it is found, i.e., it may be common only in certain areas within the unit, like bottomlands or plantations (according to FIA)?

    a.  **FIAiv** shows the average importance of the species, when it is found.
    b.  Eastern hemlock ranks 5th in overall importance across the NF (FIAsum), but 4th in importance when considering only cells where it is located—it has some particular habitats that it prefers within the NF.

4.  How abundant is each species, taken across the region of interest (according to FIA)?

    a.  **Abund** classifies **FIAsum** into: *Abundant, Common, Rare, Absent*
    b.  The Allegheny NF has 6 abundant species, 15 common species, and 22 rare species.

Appendix A.1.2. DISTRIB-II Model Outputs for Year 2100

5.  How much confidence do we have in the models?

    a.  **ModRel** gives classes of model reliability: *Low, Medium, High*
    b.  The Allegheny NF table shows the following species ratings: 18 high, 24 medium, and 15 low model reliability. Three others are labeled FIA, meaning that we only provide information from FIA, and no models were developed because too few data were available for acceptable models.

6.  What do the models suggest may happen to habitat suitability for the species by the year 2100, according to a high (or low) emissions scenario?

a. **ChngCl45** and **ChngCl85** provide, for RCP 4.5 (low emissions) and RCP 8.5 (high emissions) respectively, the potential change in suitable habitat for the species by year 2100. It is important to note that a change in suitable habitat does not mean the species importance will actually change in that area by 2100, only that the habitat is expected to increase, decrease, or remain unchanged in suitability for that species over time.

b. The Allegheny NF table shows 15 species with large or very large decrease in suitable habitat according to RCP 8.5, but also 4 species each for small decrease, small increase, or no change, 13 species for large increase, and 14 species with new habitat appearing by century's end under high emissions.

7. According to literature, how adaptable are the species to the direct and indirect impacts of a changing climate?

a. **Adapt** shows a color-coded rating (*Low—pink, Medium—yellow, High—green*) of a compilation of modification factors that estimate the adaptability of the species across their range.

b. For Allegheny NF, 15 species are classed as highly adaptable, 35 as medium, and 8 as low in adaptability.

8. How capable might the species be for coping with the changing climate, within the region of interest, under a low (or high) emissions scenario?

a. **Capabil45** (or **Capabil85**) gives an estimate of the species capability to cope, based on its change in habitat suitability (**ChngCl45** or **ChngCl85**), its **Adapt**, and its **Abund** in the region of interest. If the species is abundant locally, it is assumed to also be in refugia and niches somewhat buffered from the climate impacts. Ranks are coded *Very Good*, *Good*, *Fair*, *Poor*, *Very Poor*, *FIA only* (no model, no Capability assigned), *NNIS* (no model, non-native invasive species), *Unknown* (insufficient data to model), and *New Habitat* (potential to migrate into the region).

b. Under a scenario of RCP 8.5, the Allegheny table shows 7 species with very good and 18 with good capability, 6 with fair capability, and 3 with poor and 6 with lost capability, along with 14 with new habitat, 2 non-native invasive species, and 1 species with only FIA data.

9. What species might be considered most (or least) vulnerable to the changing climate in this area?

a. Those species listed under **Capabil45** (or **Capabil85**) as *Poor*, *Very Poor*, or *Lost* will be vulnerable according to our assessments; those listed as *Very Good*, *Good*, or *Fair* are treated as less vulnerable.

b. Most vulnerable (lost) species for the Allegheny NF under RCP 8.5 are red spruce, red pine, black ash, quaking aspen, pin cherry, and white spruce. Least vulnerable (very good) are: chestnut oak, northern red oak, white oak, white ash, black oak, eastern white pine, and American basswood. Of course, white ash is under severe threat of emerald ash borer, which the models could not adequately consider.

Appendix A.1.3. SHIFT Model Outputs for Year 2100

10. Natural migration of the species has been modeled with SHIFT—it predicts where the species may migrate within the next 100 years that is currently unoccupied by the species. Assuming a generous migration rate of ~50 km/century, where is the species likely to colonize?

a. **SHIFT45** or **SHIFT85** highlights those species that have the best chance of migrating into the region within 100 years. Those species labeled '*Migrate+*' have both new suitable habitat and some probability of colonization within 100 years, while those labeled '*Migrate++*' have a stronger potential for colonization according to the models.

b. Species with potential to migrate into the Allegheny NF within 100 years (under RCP 8.5) include: post oak, shortleaf pine, sweetgum, sycamore, and eastern redcedar (all Migrate++), flowering dogwood, sourwood, Virginia pine, and eastern redbud (all Migrate+). Under RCP 4.5, only sycamore and sourwood appear on the list.

11. What species are relatively rare in the region, but are poised to expand or infill within the region under the changing climate?

a. **SHIFT45** or **SHIFT85** labels those relatively rare species that could expand within the region as '*Infill+*' or '*Infill++*', depending on the strength of the modeled indicators.

b. The table for the Allegheny shows mockernut hickory, bitternut hickory, scarlet oak, and shagbark hickory as the best candidates for infilling (Infill++), followed by green ash and American elm (Infill+). Of course, these last two species would need to be resistant varieties to withstand current plagues against them.

12. What rare species are likely present in the region based on our model outputs, but the forest inventories missed them?

a. **SHIFT45** or **SHIFT85** labels those relatively rare species are likely present within the region as '*Likely+*' or '*Likely++*', depending on the strength of the modeled indicators.

b. The data predict that black walnut, not found in FIA plots, are now somewhere on the Allegheny NF or its buffer.

13. What species might be useful for selecting as candidates for planting in the face of a changing climate?

a. **SSO** is the species selection option to assist in decisions regarding promoting the species, where 1 indicates the species is currently present and has at least a fair capability to cope, 2 indicates the species is rare or close to the NF boundary and has a good chance of spreading inside the NF, 3 indicates the species is not recorded in FIA plots but does have some chance of getting colonized within 100 years into the NF, and 0 indicates further evaluation may be required. Generally, managers might consider selecting species first with SSO = 1, then SSO = 2, then SSO = 3, then SSO = 0.

b. For the Allegheny NF, 30 species present now have at least a fair capability to cope with the changing climate (SSO = 1); the models suggest these species are most suited for planting in the area. Three species are rare inside the NF (bitternut and mockernut hickory, slippery elm) and have potential to expand and could be suited for planting (SSO = 2) (green ash is also listed but of course would not be recommended due to emerald ash borer). Twelve additional species are marked as having some chance (>0% chance, SSO = 3) of naturally migrating there within 100 years and could be suited for planting, though with more risk of failure as compared with SSO 1 or 2. The remaining 14 species have SSO = 0, and are generally not suited for planting, though they still can be considered as candidates for planting under certain local conditions or manager preference.

14. What portion of the area has at least a 2% chance of getting colonized over 100 years? (note: these variables are only on the long file available as Supplementary File 2—Allegheny table)

a. **%2Col** estimates, for any species, the percentage of the region of interest with at least 2% chance of colonization, and is not already occupied according to FIA. One cutoff we use for potential planting guidelines is that the species requires at least 5% of the region of interest to have >2% chance of being colonized; often coded "Migrate+" or "Migrate++" under SHIFT45/SHIFT85, and coded '3' under SSO.

b.  The long form of the Allegheny table lists 49 species with at least 5% of the NF having at least a 2% chance of getting colonized over 100 years. These include species that are already present and potentially spreading to unoccupied locations within the NF.

15. What portion of the area has greater than 0% chance of getting colonized over 100 years? (note: these variables are only on the long file available as Supplementary File 2—Allegheny table)

a.  **%0Col** provides the percentage of the area, not already occupied according to FIA, that has >0% chance of getting colonized.

b.  For the Allegheny NF, 54 species (out of 60) show some area with at least some chance of getting colonized over 100 years. These also include species that are already present and potentially spreading to unoccupied locations within the NF.

16. What portion of the area has at least a 50% chance of getting colonized over 100 years? (note: these variables are only on the long file available as Supplementary File 2—Allegheny table)

a.  **%50Col** provides the percentage of the area, not already occupied according to FIA, that has >50% chance of getting colonized.

b.  Within the Allegheny, 41 species have at least some territory with >50% chance of colonization. These also include species that are already present and potentially spreading to unoccupied locations within the NF.

17. What species may have the highest (or at least some) chance of migrating and finding suitable habitat within 100 years (under low and high emissions)? (note: these variables are only on the long file available as Supplementary File 2—Allegheny table)

a.  **HQCL45** (low emissions) and **HQCL85** (high emissions) are indices that use a weighted average calculation (based on habitat quality and colonization likelihood), to assign a score for the potential to migrate into suitable habitat—the higher the value, the greater the chance, with values 1 or greater indicating the presence of both suitable and colonizable habitats.

b.  Within the Allegheny NF, of the 14 species with new habitat (question #8), 12 species have HQCL85 greater than or equal to 1.0 (indicating the presence of both suitable habitat and colonizable habitat), while for HQCL45, the number is only 3 species.

By providing answers to these 17 questions for any national forest or 1 × 1 degree grid, a user is able to delve into many aspects of their forest attributes, both now and potentially into the future. Managers and the casual user alike will have species lists showing which are abundant, common, and rare within their area of interest. Managers will have data to support (or not) their decisions related to adaptation to the changing climate. For example, selecting species that may be candidates for planting may be better justified if the HQ-CL data suggest that the suitable habitats are ripe for the species in their area of interest. This is especially true for selecting species not now identified from their area of interest–picking species with 'Migrate+' or 'Migrate++' would seem more justified in that the habitat will likely be present in coming decades and the species is close enough now to provide some chance of natural migration into the area even without assistance. Of course, any species identified via these tables must be vetted by local experts as to their suitability according to local ecological conditions. Because of the scale of analysis (8000 km$^2$ for national forests or 10,000 km$^2$ for 1 × 1 degree grids), there will always be large variations in soils, topography, land use, and hydrology within the unit that precludes or includes species as candidates. The online Climate Change Atlas website (www.fs.fed.us/atlas), as well as many other documents and web sites, also includes ecological characteristics of species to help sort out the applicability of species for particular planting sites.

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
