# Peer review of "Facilitating Adaptive Forest Management under Climate Change: A Spatially Specific Synthesis of 125 Species for Habitat Changes and Assisted Migration over the Eastern United States"

_forests, doi:10.3390/f10110989_

Round 1

Reviewer 1 Report

The originality and novelty is in the integration of the various models, inventory data, and literature synthesis – some of which has been published in the scientific literature, including this journal. What differs is the accumulated wealth of information in one manuscript, an attempt to present this information in an accessible format, and the connections to questions that managers have about climate change and resource management. However, the objective and organization of the manuscript should be clearer. The first objective in lines 45-46 describe the objective ‘to move beyond the individual evaluation of tree species … synthesize the disparate species-based results and tie them to unique geographic units, then provide a comprehensive report of current and potential future configurations of forest communities impacted by climate change. An additional objective is given on Lines 115 ‘a tabular output for a specific 1X1 or NF unit that provides species-level details on …..

If the manuscript aims to describe the synthetic approach, the manuscript does not provide sufficient information for someone to replicate this approach nor does it document why this synthetic approach is scientifically more rigorous than other currently available modeling approaches. The statistics behind the synthetic approach are impressive (analyses at different scales, rescaling across unit sizes, for example), but not described sufficiently for someone to repeat. The documentation of results and comparisons with other literature is also weak – the oaks example is valuable, interesting, and similar comparisons would strengthen the conclusion that this synthetic approach would reduce uncertainty for managers.  If the objective is to present the tables available from this integration – then a clearer description of each table that can be derived would be helpful. For example, no tables are shown for the 1X1 analysis. The first table is given for the NF analysis with the third table presenting the additional detail in a table – but the variables in Table 1 are not defined until Figure 3.     

The results are interpreted appropriately, with caveats stated on the use of models. The caveats focus on the use of the species data by managers. There are underlying assumptions associated with each model/literature synthesis. For example, the change results are based on RCP 8.5; however it is not clear if those are the results of 3 climate models described in the Iverson et al publication or only one model. The authors properly refer to the publications that preceded this integration. Are the results in this manuscript the same as in those citations or is the integration revealing differences when one takes a synthetic approach?

The two scales of analysis are of interest, but the presentation is confusing.  For example, the 1X1 grid is described first under Study Area, but the figure is in the appendix. All tables in the text are for the NF scale study. There is an explanation for mapping the 1X1 but not the NF.  The different information presented for each study area confuses the relationship of the data used – how the data moves across scale, for example.

Line comments

Terminology – the resist, adapt, migrate terminology was introduced around line 62. Then the terminology of 'infilling, likely, and migrating. And then terminology of expand. How are the terms infilling, likely, expand related to resist and adapt? 

Infilling and Likely are not defined until page 6 in the legend for Figure 3 but introduced in Table 1.

Lines 134 FIA data is discussed here. FIA sum appears in figure 3 but is defined in this paragraph as bolded element are headers on the tables. It is not clear what ‘bolded elements’ refers to – bolded elements in Figure 3, a table not yet shown?  In Figures 3, the closest term for FIAsum is FIA-SumIV, the other terms related to FIA are FIA-IV.  A careful review of terminology, use of, when it is introduced, and defined is needed. 

Line 60 – ‘tabulate them in a series of tables (1 per unit).’ It is not clear what ‘1 per unit’ means in parens with ‘series of tables’.

Table 1 – the column headings are abbreviations that need specific definitions, either as part of the table or in the text.

Line 137 – define ‘modeling cell’ –

Line 150-155. Do the breakpoints used in developing the results ChngCL45 or ChngCL85 have any ecological basis, or used in previous studies?

Table 1 - uses variables that are not defined yet. The placement of this table is a bit confusing as it is in the study area section and no mention is made or whether the same type of table can be made for the 1X1 degree grids. 

Figures 2 and 3 - it is not clear that both figures are needed, perhaps just Figure 2. Figure 3 has the variable definitions but as they are introduced earlier, perhaps the definitions should be earlier also, or table 1 comes later?

Figure 4 is valuable - what is the grid size mapped?

Figure 5 - redo the figure with all data in same size.

Figure 6 provides some information but Figures 8 and 9 are more informative relative to the topic of reducing uncertainty. 

Reviewer 2 Report

see attached file

Round 2

Reviewer 1 Report

No further suggestions

Reviewer 2 Report

My primary concerns about the originally submitted manuscript pertained to inadequate descriptions of methods. These concerns have been addressed by the authors in the new supplementary files. Reading the author responses to my original review comments, it appears that the substantive review comments have all been addressed either in the text or in the supplementary files.